# Computing Linear Restrictions of Neural Networks

**Matthew Sotoudeh**
Department of Computer Science
University of California, Davis
Davis, CA 95616
masotoudeh@ucdavis.edu

**Aditya V. Thakur**
Department of Computer Science
University of California, Davis
Davis, CA 95616
avthakur@ucdavis.edu

## Abstract

A linear restriction of a function is the same function with its domain restricted to points on a given line. This paper addresses the problem of computing a succinct representation for a linear restriction of a piecewise-linear neural network. This primitive, which we call EXACTLINE, allows us to exactly characterize the result of applying the network to all of the infinitely many points on a line. In particular, EXACTLINE computes a partitioning of the given input line segment such that the network is affine on each partition. We present an efficient algorithm for computing EXACTLINE for networks that use ReLU, MaxPool, batch normalization, fully-connected, convolutional, and other layers, along with several applications. First, we show how to exactly determine decision boundaries of an ACAS Xu neural network, providing significantly improved confidence in the results compared to prior work that sampled finitely many points in the input space. Next, we demonstrate how to exactly compute integrated gradients, which are commonly used for neural network attributions, allowing us to show that the prior heuristic-based methods had relative errors of 25-45% and show that a better sampling method can achieve higher accuracy with less computation. Finally, we use EXACTLINE to empirically falsify the core assumption behind a well-known hypothesis about adversarial examples, and in the process identify interesting properties of adversarially-trained networks.

## 1 Introduction

The past decade has seen the rise of deep neural networks (DNNs) [1] to solve a variety of problems, including image recognition [2, 3], natural-language processing [4], and autonomous vehicle control [5]. However, such models are difficult to meaningfully interpret and check for correctness. Thus, researchers have tried to understand the behavior of such networks. For instance, networks have been shown to be vulnerable to *adversarial examples*—inputs changed in a way imperceptible to humans but resulting in a misclassification by the network [6–9]–and *fooling examples*—inputs that are completely unrecognizable by humans but classified with high confidence by DNNs [10]. The presence of such adversarial and fooling inputs as well as applications in safety-critical systems has led to efforts to verify, certify, and improve robustness of DNNs [11–20]. Orthogonal approaches help visualize the behavior of the network [21–23] and interpret its decisions [24–28]. Despite the tremendous progress, more needs to be done to help understand DNNs and increase their adoption [29–32].

In this paper, we present algorithms for computing the EXACTLINE primitive: given a piecewise-linear neural network (e.g. composed of convolutional and ReLU layers) and line in the input space $\overline{QR}$, we partition $\overline{QR}$ such that the network is affine on each partition. Thus, EXACTLINE precisely captures the behavior of the network for the infinite set of points lying on the line between two points. In effect, EXACTLINE computes a succinct representation for a linear restriction of a piecewise-linear

neural network; a *linear restriction* of a function is the same function with its domain restricted to points on a given line. We present an efficient implementation of EXACTLINE (Section 2) for piecewise-linear neural networks, as well as examples of how EXACTLINE can be used to understand the behavior of DNNs. In Section 3 we consider a problem posed by Wang et al. [33], viz., to determine the classification regions of ACAS Xu [5], an aircraft collision avoidance network, when linearly interpolating between two input situations. This characterization can, for instance, determine at which precise distance from the ownship a nearby plane causes the network to instruct a hard change in direction. Section 4 describes how EXACTLINE can be used to exactly compute the *integrated gradients* [26], a state-of-the-art network attribution method that until now has only been approximated. We quantify the error of previous heuristics-based methods, and find that they result in attributions with a relative error of 25-45%. Finally, we show that a different heuristic using trapezoidal rule can produce significantly higher accuracy with fewer samples. Section 5 uses EXACTLINE to probe interesting properties of the neighborhoods around test images. We empirically reject a fundamental assumption behind the Linear Explanation of Adversarial Examples [7] on multiple networks. Finally, our results suggest that DiffAI-protected [34] neural networks exhibit significantly less non-linearity in practice, which perhaps contributes to their adversarial robustness. We have made our source code available at https://doi.org/10.5281/zenodo.3520097.

## 2  The EXACTLINE Primitive

Given a piecewise-linear neural network $f$ and two points $Q, R$ in the input space of $f$, we consider the *restriction of $f$ to $\overline{QR}$*, denoted $f_{\restriction \overline{QR}}$, which is identical to the function $f$ except that its input domain has been restricted to $\overline{QR}$. We now want to find a *succinct representation* for $f_{\restriction \overline{QR}}$ that we can analyze more readily than the neural network corresponding to $f$. In this paper, we propose to use the EXACTLINE representation, which corresponds to a *linear partitioning* of $f_{\restriction \overline{QR}}$, defined below.

**Definition 1.** *Given a function $f : A \to B$ and line segment $\overline{QR} \subseteq A$, a tuple $(P_1, P_2, P_3, \ldots, P_n)$ is a* linear partitioning *of $f_{\restriction \overline{QR}}$, denoted $\mathcal{P}\big(f_{\restriction \overline{QR}}\big)$ and referred to as "EXACTLINE of f over $\overline{QR}$," if: (1) $\{\overline{P_i P_{i+1}} \mid 1 \leq i < n\}$ partitions $\overline{QR}$ (except for overlap at endpoints); (2) $P_1 = Q$ and $P_n = R$; and (3) for all $1 \leq i < n$, there exists an affine map $A_i$ such that $f(x) = A_i(x)$ for all $x \in \overline{P_i P_{i+1}}$.*

In other words, we wish to *partition $\overline{QR}$ into a set of pieces where the action of $f$ on all points in each piece is affine.* Note that, given $\mathcal{P}\big(f_{\restriction \overline{QR}}\big) = (P_1, \ldots, P_n)$, the corresponding affine function for each partition $\overline{P_i P_{i+1}}$ can be determined by recognizing that affine maps preserve ratios along lines. In other words, given point $x = (1 - \alpha)P_i + \alpha P_{i+1}$ on linear partition $\overline{P_i P_{i+1}}$, we have $f(x) = (1 - \alpha)f(P_i) + \alpha f(P_{i+1})$. In this way, $\mathcal{P}\big(f_{\restriction \overline{QR}}\big)$ provides us a succinct and precise representation for the behavior of $f$ on all points along $\overline{QR}$.

Consider an illustrative DNN taking as input the age and income of an individual and returning a loan-approval score and premium that should be charged over a baseline amount:

$$f(X = (x_0, x_1)) = \text{ReLU}\left(A(X)\right), \text{ where } A(X) = \begin{bmatrix} -1.7 & 1.0 \\ 2.0 & -1.3 \end{bmatrix} X + \begin{bmatrix} 3 \\ 3 \end{bmatrix} \tag{1}$$

Suppose an individual of 20 years old making \$30k/year ($Q = (20, 30)$) predicts that their earnings will increase linearly every year until they reach 30 years old and are making \$50k/year ($R = (30, 50)$). We wish to understand how they will be classified by this system over these 10 years. We can use EXACTLINE (Definition 1) to compute $\mathcal{P}\big(f_{\restriction \overline{QR}}\big) = (P_1 = Q, P_2 = (23.\overline{3}, 36.\overline{6}), P_3 = (26.\overline{6}, 43.\overline{3}), P_4 = R)$, where $f_{\restriction QR}$ is *exactly* described by the following piecewise-linear function (Figure 1):

$$f_{\restriction \overline{QR}}(x) = \begin{cases} \begin{bmatrix} 0 & 0 \\ 2 & -1.3 \end{bmatrix} x + \begin{bmatrix} 0 \\ 3 \end{bmatrix}, & x \in \overline{QP_2} \\[2ex] \begin{bmatrix} -1.7 & 1 \\ 2 & -1.3 \end{bmatrix} x + \begin{bmatrix} 3 \\ 3 \end{bmatrix}, & x \in \overline{P_2 P_3} \\[2ex] \begin{bmatrix} -1.7 & 1 \\ 0 & 0 \end{bmatrix} x + \begin{bmatrix} 3 \\ 0 \end{bmatrix}, & x \in \overline{P_3 R} \end{cases} \tag{2}$$

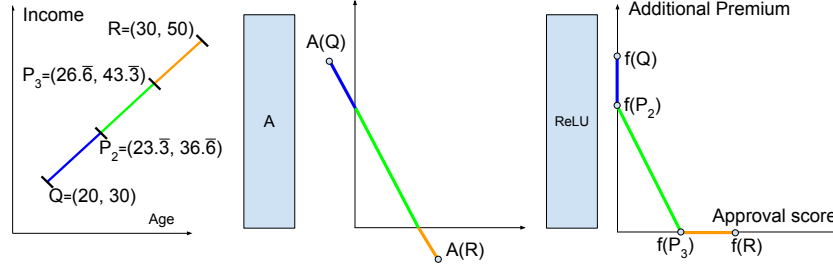

Figure 1: Computing the linear restriction of $f$ (Equation 1) using EXACTLINE. The input line segment $\overline{QR}$ is divided into three linear partitions such that the transformation from input space to output space (left plot to right plot) is affine (Equation 2). Tick marks (on the left) are used in figures throughout this paper to indicate the partition endpoints $(P_1, P_2, \ldots, P_n)$.

**Other Network Analysis Techniques**   Compared to prior work, our solution to EXACTLINE presents an interesting and unique design point in the space of neural-network analysis. Approaches such as [12, 35, 16] are precise, but exponential time because they work over the entire input domain. On another side of the design space, approaches such as those used in [15, 35, 36, 17, 37, 33] are significantly faster while still working over the full-dimensional input space, but accomplish this by trading analysis precision for speed. This trade-off between speed and precision is particularly well-illuminated by [38], which monotonically refines its analysis when given more time. In contrast, the key observation underlying our work is that we can perform *both* an efficient (worst-case polynomial time for a fixed number of layers) and precise analysis by *restricting the input domain* to be one dimensional (a line). This insight opens a new dimension to the discussion of network analysis tools, showing that *dimensionality* can be traded for significant gains in *both* precision and efficiency (as opposed to prior work which has explored the tradeoff primarily along the precision and efficiency axes under the assumption of high-dimensional input regions). Hanin and Rolnick [39] similarly considers one-dimensional input spaces, but the paper is focused on a number of theoretical properties and does not focus on the algorithm used in their empirical results.

**Algorithm**   We will first discuss computation of EXACTLINE for individual layers. Note that by definition, EXACTLINE for affine layers does not introduce any new linear partitions. This is captured by Theorem 1 (proved in Appendix D) below:

**Theorem 1.** *For any affine function $A : X \to Y$ and line segment $\overline{QR} \subset X$, the following is a suitable linear partitioning (Definition 1): $\mathcal{P}\left(A_{\restriction\overline{QR}}\right) = (Q, R)$.*

The following theorem (proved in Appendix E) presents a method of computing $\mathcal{P}\left(\mathrm{ReLU}_{\restriction\overline{QR}}\right)$.

**Theorem 2.** *Given a line segment $\overline{QR}$ in $d$ dimensions and a rectified linear layer $\mathrm{ReLU}(x) = (\max(x_1, 0), \ldots, \max(x_d, 0))$, the following is a suitable linear partitioning (Definition 1):*

$$\mathcal{P}\left(\mathrm{ReLU}_{\restriction\overline{QR}}\right) = \mathrm{sorted}\left(\left(\{Q, R\} \cup \{Q + \alpha(R - Q) \mid \alpha \in D\}\right) \cap \overline{QR}\right), \qquad (3)$$

*where $D = \{-Q_i/(R_i - Q_i) \mid 1 \le i \le d\}$, $V_i$ is the ith component of vector $V$, and $\mathrm{sorted}$ returns a tuple of the points sorted by distance from $Q$.*

The essential insight is that we can "follow" the line until an orthant boundary is reached, at which point a new linear region begins. To that end, each number in $D$ represents a ratio between $Q$ and $R$ at which $\overline{QR}$ crosses an orthant boundary. Notably, $D$ actually computes such "crossing ratios" for the *unbounded* line $\overline{QR}$, hence intersecting the generated endpoints with $\overline{QR}$ in Equation 3.

An analogous algorithm for MaxPool is presented in Appendix F; the intuition is to follow the line until the maximum in any window changes. When a ReLU layer is followed by a MaxPool layer (or vice-versa), the "fused" algorithm described in Appendix G can improve efficiency significantly. More generally, the algorithm described in Appendix H can compute EXACTLINE for any piecewise-linear function.

Finally, in practice we want to compute $\mathcal{P}\left(f_{\restriction\overline{QR}}\right)$ for entire *neural networks* (i.e. sequential compositions of layers), not just individual layers (as we have demonstrated above). The next theorem

shows that, as long as one can compute $\mathcal{P}\left(L_{i\restriction\overline{MN}}\right)$ for each individual layer $L_i$ and arbitrary line segment $\overline{MN}$, then these algorithms can be *composed* to compute $\mathcal{P}\left(f_{\restriction\overline{QR}}\right)$ *for the entire network*.

**Theorem 3.** *Given any piecewise-linear functions $f, g, h$ such that $f = h \circ g$ along with a line segment $\overline{QR}$ where $g(R) \neq g(Q)$ and $\mathcal{P}\left(g_{\restriction\overline{QR}}\right) = (P_1, P_2, \ldots, P_n)$ is* EXACTLINE *applied to $g$ over $\overline{QR}$, the following holds:*

$$\mathcal{P}\left(f_{\restriction\overline{QR}}\right) = \text{sorted}\left(\bigcup_{i=1}^{n-1}\left\{P_i + \frac{y - g(P_i)}{g(P_{i+1}) - g(P_i)} \times (P_{i+1} - P_i) \mid y \in \mathcal{P}\left(h_{\restriction\overline{g(P_i)g(P_{i+1})}}\right)\right\}\right)$$

*where* sorted *returns a tuple of the points sorted by distance from Q.*

The key insight is that we can first compute EXACTLINE for the first layer, i.e. $\mathcal{P}\left(L_{1\restriction\overline{QR}}\right) = (P_1^1, P_2^1, \ldots, P_n^1)$, then we can continue computing EXACTLINE for the rest of the network *within each of the partitions $\overline{P_i^1 P_{i+1}^1}$ individually.*

In Appendix C we show that, over arbitrarily many affine layers, $l$ ReLU layers each with $d$ units, and $m$ MaxPool or MaxPool + ReLU layers with $w$ windows each of size $s$, at most $\text{O}((d + ws)^{l+m})$ segments may be produced. If only ReLU and affine layers are used, at most $\text{O}(d^l)$ segments may be produced. Notably, this is a significant improvement over the $\text{O}((2^d)^l)$ upper-bound and $\Omega(l \cdot (2^d))$ lower-bound of Xiang et al. [35]. One major reason for our improvement is that we particularly consider one-dimensional input *lines* as opposed to arbitrary polytopes. Lines represent a particularly efficient special case as they are efficiently representable (by their endpoints) and, being one-dimensional, are not subject to the combinatorial blow-up faced by transforming larger input regions. Furthermore, in practice, we have found that the majority of ReLU nodes are "stable", and the actual number of segments remains tractable; this algorithm for EXACTLINE often executes in a matter of seconds for networks with over $60,000$ units (whereas the algorithm of Xiang et al. [35] runs in at least exponential $\text{O}(l \cdot (2^d))$ time regardless of the input region as it relies on trivially considering *all possible* orthants).

## 3    Characterizing Decision Boundaries for ACAS Xu

The first application of EXACTLINE we consider is that of understanding the decision boundaries of a neural network over some infinite set of inputs. As a motivating example, we consider the ACAS Xu network trained by Julian et al. [5] to determine what action an aircraft (the "ownship") should take in order to avoid a collision with an intruder. After training such a network, one usually wishes to probe and visualize the recommendations of the network. This is desirable, for example, to determine at what distance from the ownship an intruder causes the system to suggest a strong change in heading, or to ensure that distance is roughly the same regardless of which side the intruder approaches.

The simplest approach, shown in Figure 2f and currently the standard in prior work, is to consider a (finite) set of possible input situations (samples) and see how the network reacts to each of them. This can help one get an overall idea of how the network behaves. For example, in Figure 2f, we can see that the network has a mostly symmetric output, usually advising the plane to turn away from the intruder when sufficiently close. Although sampling in this way gives human viewers an intuitive and meaningful way of understanding the network's behavior, it is severely limited because it relies on sampling *finitely many points* from a (practically) *infinite input space*. Thus, there is a significant chance that some interesting or dangerous behavior of the network may be exposed with more samples.

By contrast, the EXACTLINE primitive can be used to *exactly* determine the output of the network at *all* of the *infinitely many* points on a line in the input region. For example, in Figure 2a, we have used EXACTLINE to visualize a particular head-on collision scenario where we vary the distance of the intruder (specified in polar coordinates $(\rho, \theta)$) with respect to the ownship (always at $(0, 0)$). Notably, there is a region of "Strong Left" in a region of the line that is otherwise entirely "Weak Left" that shows up in Figure 2a (the EXACTLINE method) but not in Figure 2b (the sampling method). We can do this for lines varying the $\theta$ parameter instead of $\rho$, result in Figure 2c and Figure 2d. Finally, repeating this process for many lines and overlapping them on the same graph produces a detailed "grid" as shown in Figure 2e.

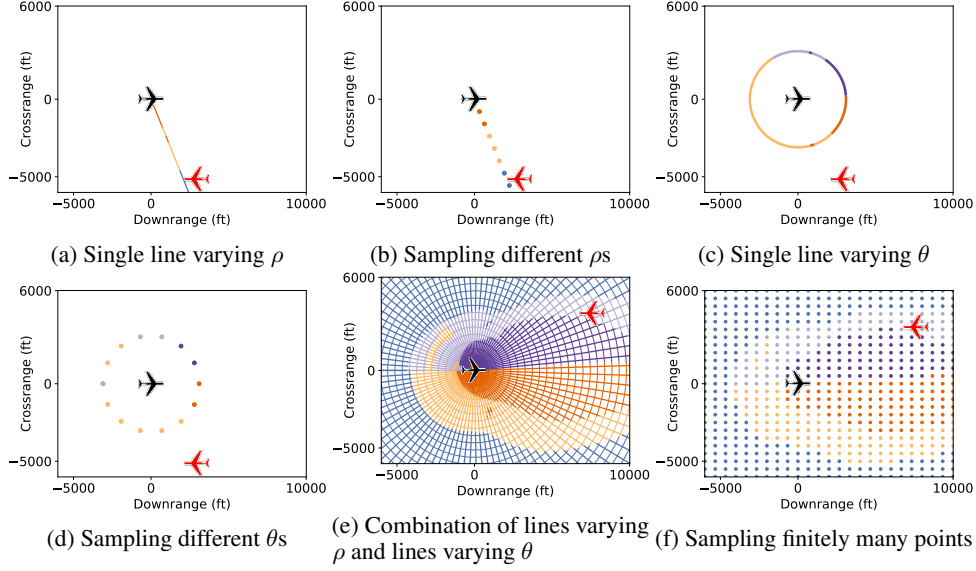

Legend: ━ Clear-of-Conflict, ━ Weak Right, ━ Strong Right, ━ Strong Left, ━ Weak Left.

(a) Single line varying $\rho$

(b) Sampling different $\rho$s

(c) Single line varying $\theta$

(d) Sampling different $\theta$s

(e) Combination of lines varying $\rho$ and lines varying $\theta$

(f) Sampling finitely many points

Figure 2: (a)–(d) Understanding the decision boundary of an ACAS Xu aircraft avoidance network along individual lines using EXACTLINE ((a), (c)) and finite sampling ((b), (d)). In the EXACTLINE visualizations there is a clear region of "strong left" in a region that is otherwise "weak left" that does not show in the sampling plots due to the sampling density chosen. In practice, it is not clear what sampling density to choose, thus the resulting plots can be inaccurate and/or misleading. (e)–(f) Computing the decision boundaries among multiple lines and plotting on the same graph. Using EXACTLINE to sample *infinitely many* points provides more confidence in the interpretation of the decision boundaries. Compare to similar figures in Julian et al. [5], Katz et al. [12].

Figure 2e also shows a number of interesting and potentially dangerous behaviors. For example, there is a significant region behind the plane where an intruder on the left may cause the ownship to make a weak left turn *towards* the intruder, an unexpected and asymmetric behavior. Furthermore, there are clear regions of strong left/right where the network otherwise advises weak left/right. Meanwhile, in Figure 2f, we see that the sampling density used is too low to notice the majority of this behavior. In practice, it is not clear what sampling density should be taken to ensure all potentially-dangerous behaviors are caught, which is unacceptable for safety-critical systems such as aircraft collision avoidance.

**Takeaways.** EXACTLINE can be used to visualize the network's output on *infinite subsets of the input space*, significantly improving the confidence one can have in the resulting visualization and in the safety and accuracy of the model being visualized.

**Future Work.** One particular area of future work in this direction is using EXACTLINE to assist in network verification tools such as Katz et al. [12] and Gehr et al. [15]. For example, the relatively-fast EXACTLINE could be used to check infinite subsets of the input space for counter-examples (which can then be returned immediately) before calling the more-expensive complete verification tools.

## 4 Exact Computation of Integrated Gradients

Integrated Gradients (IG) [26] is a method of attributing the prediction of a DNN to its input features. Suppose function $F : \mathbb{R}^n \rightarrow [0, 1]$ defines the network. The *integrated gradient* along the $i^{th}$ dimension for an input $x = (x_1, \ldots, x_n) \in \mathbb{R}^n$ and baseline $x' \in \mathbb{R}^n$ is defined as:

$$IG_i(x) \stackrel{\text{def}}{=} (x_i - x'_i) \times \int_{\alpha=0}^{1} \frac{\partial F(x' + \alpha \times (x - x'))}{\partial x_i} d\alpha \qquad (4)$$

Thus, the integrated gradient along all input dimensions is the integral of the gradient computed on all points on the straightline path from the baseline $x'$ to the input $x$. In prior work it was not known how

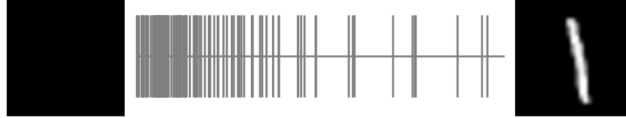

Figure 3: "Integrated Gradients" is a powerful method of neural network attribution. IG relies on computing the integral of the gradients of the network at all points linearly interpolated between two images (as shown above), however previous work has only been able to *approximate* the true IG, casting uncertainty on the results. Within each partition identified by EXACTLINE (delineated by vertical lines) the gradient is constant, so computing the *exact* IG is possible for the first time.

to exactly compute this integral for complex networks, so it was approximated using the left-Riemann summation of the gradients at $m$ uniformly sampled points along the straightline path from $x'$ to $x$:

$$\widetilde{IG}_i^m \overset{\text{def}}{=} (x_i - x_i') \times \sum_{0 \leq k < m} \frac{\partial F(x' + \frac{k}{m} \times (x - x'))}{\partial x_i} \times \frac{1}{m} \tag{5}$$

The number of samples $m$ determines the quality of this approximation. Let $\widetilde{m}$ denote the number of samples that is large enough to ensure that $\sum_{i=1}^n \widetilde{IG}_i^{\widetilde{m}} \approx F(x) - F(x')$. This is the recommended number of samples suggested by Sundararajan et al. [26]. In practice, $\widetilde{m}$ can range between 20 and 1000 [26, 40].

While the (exact) IG described by Equation 4 satisfies properties such as completeness [26, §3] and sensitivity [26, §2.1], the approximation computed in practice using Equation 5 need not.

The integral in Equation 4 can be computed exactly by adding an additional condition to the definition of EXACTLINE: that the gradient of the network within each partition is constant. It turns out that this stricter definition is met by all of the algorithms we have discussed so far, a fact we discuss in more detail in Appendix J. For ReLU layers, for example, because the network acts like a single affine map within each orthant, and we split the line such that each partition is entirely contained within an orthant, the network's gradient is constant within each orthant (and thus along each EXACTLINE partition). This is stated formally by Theorem 4 and proved in Appendix J:

**Theorem 4.** *For any network $f$ with nonlinearities introduced only by ReLU functions and $\mathcal{P}\big(f_{\restriction \overline{QR}}\big) = (P_1, P_2, \ldots, P_n)$ computed according to Equation 3, the gradient of $f$ with respect to its input vector $x$, i.e. $\nabla f(x)$, is constant within each linear partition $\overline{P_i P_{i+1}}$.*

This allows us to exactly compute the IG of each individual partition $r$ ($RIG_i^r$) by finding the gradient at any point in that partition and multiplying by the width of the partition:

$$RIG_i^r(x) \overset{\text{def}}{=} (P_{(r+1)i} - P_{ri}) \times \frac{\partial F(0.5 \times (P_{ri} + P_{(r+1)i}))}{\partial x_i} \tag{6}$$

Compared to Equation 4, Equation 6 computes the IG for partition for $\overline{P_r P_{r+1}}$ by replacing the integral with a single term (arbitrarily choosing $\alpha = 0.5$, i.e. the midpoint) because the gradient is uniform along $\overline{P_r P_{r+1}}$. The exact IG of $\overline{x'x}$ is the sum of the IGs of each partition:

$$IG_i(x) = \sum_{r=1}^n RIG_i^r(x) \tag{7}$$

**Empirical Results.** A prime interest of ours was to determine the accuracy of the existing sampling method. On three different CIFAR10 networks [41], we took each image in the test set and computed the exact IG against a black baseline using Equation 7. We then found $\widetilde{m}$ and computed the mean relative error between the exact IG and the approximate one. As shown in Table 1, the approximate IG has an error of $25 - 45\%$. This is a concerning result, indicating that the existing use of IG in practice may be misleading. Notably, without EXACTLINE, there would be no way of realizing this issue, as this analysis relies on computing the exact IG to compare with.

For many smaller networks considered, we have found that computing the exact IG is relatively fast (i.e., at most a few seconds) and would recommend doing so for situations where it is feasible. However, in some cases the number of gradients that would have to be taken to compute exact IG is high (see Column 2, Table 2). In such cases, we can use our exact computation to understand how

Table 1: Mean relative error for approximate IG (using $\widetilde{m}$) compared to exact IG (using EXACTLINE) on CIFAR10. The approximation error is surprisingly high.

| | Error (%) |
|---|---|
| convsmall | 24.95 |
| convmedium | 24.05 |
| convbig | 45.34 |

Table 2: Average number of samples needed by different IG approximations to reach $5\%$ relative error w.r.t. exact IG (using EXACTLINE). Outliers requiring over $1,000$ samples were not considered. Using trapezoidal rule instead of a left-sum can cause large gains in accuracy and performance.

| | Exact | Approximate | | |
|---|---|---|---|---|
| | | left | right | trap. |
| convsmall | 2582.74 | 136.74 | 139.40 | 91.67 |
| convmedium | 3578.89 | 150.31 | 147.59 | 91.57 |
| convbig | 14064.65 | 269.43 | 278.20 | 222.79 |

many uniform samples one should use to compute the approximate IG to within $5\%$ of the exact IG. To that end, we performed a second experiment by increasing the number of samples taken until the approximate IG reached within $5\%$ error of the exact IG. In practice, we found there were a number of "lucky guess" situations where taking, for example, exactly 5 samples led to very high accuracy, but taking 4 or 6 resulted in very poor accuracy. To minimize the impact of such outliers, we also require that the result is within $5\%$ relative error when taking up to 5 more samples. In Table 2 the results of this experiment are shown under the column "left," where it becomes clear that relatively few samples (compared to the number of actual linear regions under "exact") are actually needed. Thus, one can use EXACTLINE on a test set to understand how many samples are needed on average to get within a desired accuracy, then use that many samples when approximating IG.

Finally, the left Riemann sum used by existing work in IG [26] is only one of many possible sampling methods; one could also use a right Riemann sum or the "Trapezoidal Rule." With EXACTLINE we can compute the exact IG and quantify how much better or worse each approximation method is. To that end, the columns "right" and "trap." in Table 2 show the number of samples one must take to get consistently within $5\%$ of the exact integrated gradients. Our results show the number of samples needed to accurately approximate IG with trapezoidal rule is significantly less than using left and right sums. Note that, because these networks are piecewise-linear functions, it is possible for trapezoidal sampling to be worse than left or right sampling, and in fact for all tested models there were images where trapezoidal sampling was less accurate than left and right sampling. This is why having the ability to compute the *exact* IG is important, to ensure that we can empirically quantify the error involved in and justify choices between different heuristics. Furthermore, we find a per-image reduction in the number of samples needed of $20 - 40\%$ on average. Thus, we also recommend that users of IG use a trapezoidal approximation instead of the currently-standard left sum.

**Takeaways.** EXACTLINE is the first method for exactly computing integrated gradients, as all other uses of the technique have sampled according to Equation 5. From our results, we provide two suggestions to practitioners looking to use IG: (1) Use EXACTLINE on a test set before deployment to better understand how the number of samples taken relates to approximation error, then choose sampling densities at runtime based on those statistics and the desired accuracy; (2) Use the trapezoidal rule approximation when approximating to get better accuracy with fewer samples.

**Future Work.** EXACTLINE can be used to design new IG approximation using non-uniform sampling. EXACTLINE can similarly be used to exactly compute other measures involving path integrals such as neuron conductance [42], a refinement of integrated gradients.

## 5 Understanding Adversarial Examples

We use EXACTLINE to empirically investigate and falsify a fundamental assumption behind a well-known hypothesis for the existence of adversarial examples. In the process, using EXACTLINE, we discover a strong association between robustness and the empirical *linearity* of a network, which we believe may help spur future work in understanding the source of adversarial examples.

**Empirically Testing the Linear Explanation of Adversarial Examples.** We consider in this section one of the dominant hypotheses for the problem of adversarial examples ([6–8]), first proposed by Goodfellow et al. [7] and termed the "Linear Explanation of Adversarial Examples." It makes a fundamental assumption that, when restricted to the region around a particular input point, the

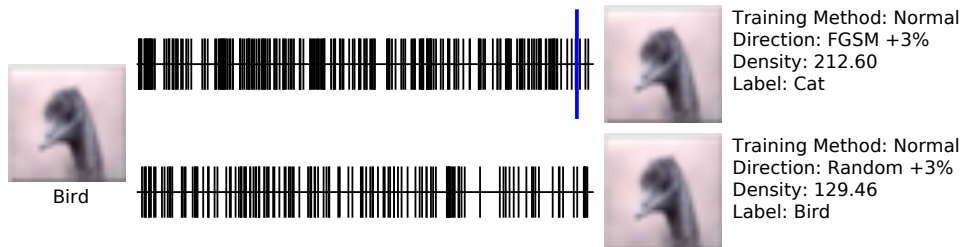

Figure 4: Comparing the density of linear partitions from a test image to FGSM and random baselines. The long blue line segment indicates the change in classification. (1) Even within a small perturbation of the input point, there is *significant* non-linear behavior in the network. (2) Although the linear hypothesis for adversarial examples predicts that both normal inputs and their corresponding adversarial examples ("FGSM + 3%") will lie on the same linear region, we find in practice that, not only do they lie on different linear regions, but there is significantly *more* non-linearity in the adversarial (FGSM) direction than a random direction. This falsifies the fundamental assumption behind the linear explanation of adversarial examples.

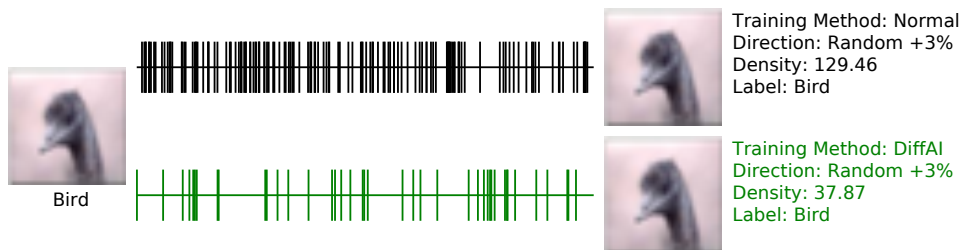

Figure 5: Comparing the density of linear partitions from a test image to random baselines for normal (in black) and DiffAI-protected networks (in green). Networks trained with the DiffAI robust-training scheme tend to exhibit significantly fewer non-linearities than their normally-trained counterparts.

output of the neural network network is *linear*, i.e. the tangent plane to the network's output at that point exactly matches the network's true output. Starting from this assumption, Goodfellow et al. [7] makes a theoretical argument concluding that the phenomenon of adversarial examples results from the network being "too linear." Although theoretical debate has ensued as to the merits of this argument [43], the effectiveness of adversarials generated according to this hypothesis (i.e., using the standard "fast gradient sign method" or "FGSM") has sustained it as one of the most well-subscribed-to theories as to the source of adversarial examples. However, to our knowledge, the fundamental assumption of the theory—that real images and their adversarial counterparts lie on the same linear region of a neural network—has not been rigorously validated empirically.

To empirically validate this hypothesis, we looked at the line between an image and its FGSM-perturbed counterpart (which is classified differently by the network), then used EXACTLINE to quantify the *density* of each line (the number of linear partitions — each delineated by tick marks in Figure 4 — divided by the distance between the endpoints). If the underlying linearity assumption holds, we would expect that both the real image and the perturbed image lie on the same linear partition, thus we would not expect to see the line between them pass through any other linear partitions. In fact (top of Figure 4), we find that the adversarial image seems to lie across many linear partitions, directly contradicting the fundamental assumption of Goodfellow et al. [7]. We also compared this line to the line between the real image and the real image *randomly* perturbed by the same amount (bottom of Figure 4). As shown in Table 3, the FGSM direction seems to pass through significantly *more* linear partitions than a randomly chosen direction. This result shows that, not only is the linearity assumption not met, but in fact the opposite appears to be true: adversarial examples are associated with *unusually non-linear* directions of the network.

With these results in mind, we realized that the linearity assumption as initially presented in Goodfellow et al. [7] is stronger than necessary; it need only be the case that the gradient is *reasonably* constant across the line, so that the tangent plane approximation used is still reasonably accurate. To

Table 3: Density of FGSM line partitions divided by density of Random line partitions. FGSM directions tend to be significantly more dense than random directions, contradicting the well-known Linear Explanation of Adversarial Examples. Mdn: Median, 25%: 25% percentile, 75%: 75% percentile

(a) MNIST

| Training Method | FGSM/Random | | |
|---|---|---|---|
| | Mdn | 25% | 75% |
| Normal | 1.36 | 0.99 | 1.76 |
| DiffAI | 0.98 | 0.92 | 1.38 |
| PGD | 1.22 | 0.97 | 1.51 |

(b) CIFAR10

| Training Method | FGSM/Random | | |
|---|---|---|---|
| | Mdn | 25% | 75% |
| Normal | 1.78 | 1.60 | 2.02 |
| DiffAI | 1.67 | 1.47 | 2.03 |
| PGD | 1.84 | 1.65 | 2.10 |

Table 4: Comparing density of lines when different training algorithms (normal, DiffAI, and PGD) are used. We report the mean of those ratios across all tested lines. These results indicate that networks trained to be adversarially robust with DiffAI or PGD training methods tend to behave more linearly than non-robust models. Mdn: Median, 25%: 25% percentile, 75%: 75% percentile

(a) MNIST

| Dir. | Normal/DiffAI | | | Normal/PGD | | |
|---|---|---|---|---|---|---|
| | Mdn | 25% | 75% | Mdn | 25% | 75% |
| FGSM | 3.05 | 2.05 | 4.11 | 0.88 | 0.66 | 1.14 |
| Rand. | 2.50 | 1.67 | 3.00 | 0.80 | 0.62 | 1.00 |

(b) CIFAR10

| Dir. | Normal/DiffAI | | | Normal/PGD | | |
|---|---|---|---|---|---|---|
| | Mdn | 25% | 75% | Mdn | 25% | 75% |
| FGSM | 3.37 | 2.77 | 4.94 | 1.48 | 1.22 | 1.67 |
| Rand. | 3.43 | 2.78 | 4.42 | 1.51 | 1.21 | 1.80 |

measure how well the tangent plane approximates the function between normal and adversarial inputs, we compared the gradient taken at the real image (i.e., used by FGSM) to the gradients at each of the intermediate linear partitions, finding the mean error between each and averaging weighted by size of the partition. If this number is near $0$, it implies that, although many theoretically distinct linear partitions exist between the two points, they have roughly the same "slope" and thus the tangent plane approximation would be accurate and the Linearity Hypothesis may still be a worthwhile explanation of the phenomenon of adversarial examples. However, we find that this number is larger than $250\%$ when averaged over all images tested, implying that the tangent plane is *not* a particularly good approximation of the underlying function in these regions of input space and providing further empirical evidence against the Linear Explanation of adversarial examples.

**Characteristics of Adversarially-Trained Networks.** We also noticed an unexpected trend in the previous experiment: networks trained to be robust to adversarial perturbations (particularly DiffAI-trained networks [34]) seemed to have *significantly* fewer linear partitions in all of the lines that we evaluated them on (see Figure 5). Further experiments, summarized in Table 4, showed that networks trained to be adversarially robust with PGD and especially DiffAI training methods exhibit up to $5\times$ fewer linear partitions for the same change in input. This observation suggests that the neighborhoods around points in adversarially-trained networks are "flat" (more linear).

**Takeaways.** Our results falsify the fundamental assumption behind the well-known Linear Explanation for adversarial examples. Adversarial training tends to make networks more linear.

**Future Work.** EXACTLINE can be used to investigate adversarial robustness. Further investigation into why DiffAI-protected networks tend to be more linear will help resolve the question (raised in this work) of whether reduced density of linear partitions contributes to robustness, or increased robustness results in fewer linear partitions (or if there is a third important variable impacting both).

# 6 Conclusion

We address the problem of computing a succinct representation of a linear restriction of a neural network. We presented EXACTLINE, a novel primitive for the analysis of piecewise-linear deep neural networks. Our algorithm runs in a matter of a few seconds on large convolutional and ReLU networks, including ACAS Xu, MNIST, and CIFAR10. This allows us to investigate questions about these networks, both shedding new light and raising new questions about their behavior.

**Acknowledgements**

We thank Nina Amenta, Yong Jae Lee, Cindy Rubio-González, and Mukund Sundararajan for their feedback and suggestions on this work. This material is based upon work supported by AWS Cloud Credits for Research.

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
