[Supplementary Material]

# Computing Linear Restrictions of Neural Networks: Supplemental

## A    Specification of Evaluation Hardware

Although we do not claim particular performance results, we do point out that all EXACTLINE uses in our experiments took only a matter of seconds on commodity hardware (although in some cases the experiments themselves took a few minutes, particularly when computing gradients for Section 4).

For reproducibility, all experimental data reported was run on Amazon EC2 c5.metal instances, using BenchExec [44] to limit to 16 CPU cores and 16 GB of memory. We have also run the results on commodity hardware, namely an Intel Core i7-7820X CPU at 3.6GHz with 32GB of memory (both resources shared with others simultaneously), for which the "matter of seconds" characterization above holds as well.

All experiments were only run on CPU, although we believe computing EXACTLINE on GPUs is an important direction for future research on significantly larger networks.

## B    Uniqueness of EXACTLINE

The *smallest* tuple satisfying the requirements on $\mathcal{P}\big(f_{\restriction\overline{QR}}\big)$ is unique (when it exists) for any given $f$ and $QR$, and any tuple satisfying $\mathcal{P}\big(f_{\restriction\overline{QR}}\big)$ can be converted to this minimal tuple by removing any endpoint $P_i$ which lies on $\overline{P_{i-1}P_{i+1}}$. In the proceeding text we will discuss methods for computing *some* tuple satisfying $\mathcal{P}\big(f_{\restriction\overline{QR}}\big)$; if the reader desires, they can use the procedure mentioned in the previous sentence to reduce this to the unique smallest such tuple. However, we note that the algorithms below *usually* produce the minimal tuple on real-world networks even without any reduction procedure due to the high dimensionality of the functions involved.

## C    Runtime of EXACTLINE Algorithms

We note that the algorithm corresponding to Theorem 1 runs on a single line segment in constant time producing a single resulting line segment. The algorithm corresponding to Equation 3 runs on a single line segment in $\mathrm{O}(d)$ time, producing at most $\mathrm{O}(d)$ new segments. If $w$ is the number of windows and $s$ is the size of each window, the algorithms for MaxPool and ReLU + MaxPool run in time $\mathrm{O}(ws^2)$ and produce $\mathrm{O}(ws)$ new line segments. Thus, using the algorithm corresponding to Theorem 3, over arbitrarily many affine layers, $l$ ReLU layers each with $d$ units, and $m$ MaxPool or MaxPool + ReLU layers with $w$ windows each of size $s$, then at most $\mathrm{O}((d + ws)^{l+m})$ segments may be produced. If only $l$ ReLU and arbitrarily-many affine layers are used, at most $\mathrm{O}(d^l)$ segments may be produced.

## D    EXACTLINE for Affine Layers

**Theorem 1.** *For any affine function $A : X \to Y$ and line segment $\overline{QR} \subset X$, the following is a suitable linear partitioning (Definition 1): $\mathcal{P}\big(A_{\restriction\overline{QR}}\big) = (Q, R)$.*

*Proof.* By the definition of $\mathcal{P}\big(A_{\restriction\overline{QR}}\big)$ it suffices to show that $\{\overline{QR}\}$ partitions $\overline{QR}$ and produce an affine map $A'$ such that $A(x) = A'(x)$ for every $x \in \overline{QR}$.

The first fact follows directly, as $\overline{QR} = \overline{QR} \implies \overline{QR} \subseteq \overline{QR}$ and every element of $\overline{QR}$ belongs to $\overline{QR}$.

For the second requirement, we claim that $A' = A$ satisfies the desired property, as $A$ is affine and $A(x) = A(x)$ for all $x$ in general and in particular for all $x \in \overline{QR}$.     □

# E EXACTLINE for ReLU Layers

**Theorem 2.** *Given a line segment $\overline{QR}$ in $d$ dimensions and a rectified linear layer $\mathrm{ReLU}(x) = (\max(x_1, 0), \ldots, \max(x_d, 0))$, the following is a suitable linear partitioning (Definition 1):*

$$\mathcal{P}\big(\mathrm{ReLU}_{\restriction \overline{QR}}\big) = \mathrm{sorted}\left(\left(\{Q, R\} \cup \{Q + \alpha(R - Q) \mid \alpha \in D\}\right) \cap \overline{QR}\right), \qquad (3)$$

*where $D = \{-Q_i/(R_i - Q_i) \mid 1 \leq i \leq d\}$, $V_i$ is the $i$th component of vector $V$, and* sorted *returns a tuple of the points sorted by distance from $Q$.*

*Proof.* First, we define the ReLU function like so:

$$\begin{bmatrix} \delta_{1\mathrm{sign}(x_1)} & 0 & \cdots & 0 \\ 0 & \delta_{1\mathrm{sign}(x_2)} & \cdots & 0 \\ \vdots & \vdots & \ddots & \vdots \\ 0 & 0 & \cdots & \delta_{1\mathrm{sign}(x_d)} \end{bmatrix} \begin{bmatrix} x_1 \\ x_2 \\ \vdots \\ x_d \end{bmatrix}$$

where $\mathrm{sign}(x)$ returns 1 if $x$ is positive and 0 otherwise while $\delta_{ij}$ is the Kronecker delta.

Now, it becomes clear that, as long as the signs of each $x_i$ are constant, the ReLU function is linear.

We note that, over a Euclidean line segment $\overline{QR}$, we can parameterize $\overline{QR}$ as $\overline{QR}(\alpha) = Q + \alpha(R - Q)$. Considering the $i$th component, we have a linear relation $\overline{QR}_i(\alpha) = Q_i + \alpha(R_i - Q_i)$ which changes sign at most once, when $\overline{QR}_i(\alpha) = 0$ (because linear functions in $\mathbb{R}$ are continuous and monotonic). Thus, we can solve for the sign change of the $i$th dimension as:

$$\overline{QR}_i(\alpha) = 0$$
$$\implies 0 = Q_i + \alpha(R_i - Q_i)$$
$$\implies \alpha = -\frac{Q_i}{R_i - Q_i}$$

As we have restricted the function to $0 \leq \alpha \leq 1$, at any $\alpha$ within these bounds the sign of some component changes and the function acts non-linearly. Between any two such $\alpha$s, however, the signs of all components are constant, so the ReLU function is perfectly described by a linear map as shown above. Finally, we can solve for the endpoints corresponding to any such $\alpha$ using the parameterization $\overline{QR}(\alpha)$ defined above, resulting in the formula in the theorem.

$Q, R$ are included to meet the partitioning definition, as the sign of some element may not be 0 at the $Q, R$ endpoints. $\qquad \square$

# F EXACTLINE for MaxPool Layers

As discussed in the paper, although we do not use MaxPool layers in any of our evaluated networks, we have developed and implemented an algorithm for computing $\mathcal{P}\big(\mathrm{MaxPool}_{\restriction \overline{QR}}\big)$, which we present here. In particular, we present $\mathcal{P}\big(\mathrm{MaxPoolWindow}_{\restriction \overline{QR}}\big)$, i.e. the linear restriction for any given window. $\mathcal{P}\big(\mathrm{MaxPool}_{\restriction \overline{MN}}\big)$ can be then be computed by separating each window $\overline{QR}$ from

$\overline{MN}$ and applying $\mathcal{P}\big(\text{MaxPoolWindow}_{\restriction \overline{QR}}\big)$. Notably, there may be duplicate endpoints (e.g. if there is overlap in the windows) which can be handled by removing duplicates if desired.

---

**Algorithm 1:** $\mathcal{P}\big(\text{MaxPoolWindow}_{\restriction \overline{QR}}\big)$. Binary operations involving both scalars and vectors apply the operation element-wise to each component of the vector.

---

**Input:** $\overline{QR}$, the line segment to restrict the MaxPoolWindow function to.
**Output:** $\mathcal{P}\big(\text{MaxPoolWindow}_{\restriction \overline{QR}}\big)$

1   $\mathcal{P} \leftarrow [Q]$   `// Begin an (ordered) list of points with one item, `$Q$`.`
2   $\alpha \leftarrow 0.0$   `// Ratio along `$\overline{QR}$` of the last endpoint in `$\mathcal{P}$`.`
3   $m \leftarrow \text{argmax}(Q)$   `// Maximum component of the last endpoint in `$\mathcal{P}$`.`
4   **while** $m \neq \text{argmax}(R)$ **do**
5      $D \leftarrow \frac{Q - Q_m}{(R_i - Q_i) - (R - Q)}$
6      $A \leftarrow \{(D_i, i) \mid 1 \leq i \leq d \wedge \alpha < D_i < 1.0\}$
7      **if** $A = \emptyset$ **then break**
8      $(\alpha, m) \leftarrow \text{lexmin}(A)$   `// Lexicographical minimum of the tuples in `$A$`.`
9      $\text{append}(\mathcal{P}, Q + \alpha \times (R - Q))$
10   $\text{append}(\mathcal{P}, R)$
11   **return** $\mathcal{P}$   `// Interpret the list `$\mathcal{P}$` as a tuple and return it.`

---

*Proof.* MaxPool applies a separate mapping from each input window to each output component, so it suffices to consider each window separately.

Within a given window, the MaxPool operation returns the value of the maximum component. It is thus linear while the index of the maximum component remains constant. Now, we parameterize $\overline{QR}(\alpha) = Q + \alpha \times (R - Q)$. At each iteration of the loop we solve for the next point at which the maximum index changes. Assuming the maximum index is $m$ when $\alpha = \alpha_m$, we can solve for the next ratio $\alpha_i > \alpha_m$ at which index $i$ will become larger than $m$ like so (again realizing that linear functions are monotonic):

$$\overline{QR}_m(\alpha_i) = \overline{QR}_i(\alpha_i)$$
$$\implies Q_m + \alpha_i \times (R_m - Q_m) = Q_i + \alpha_i \times (R_i - Q_i)$$
$$\implies \alpha_i \times (R_m - Q_m + Q_i - R_i) = Q_i - Q_m$$
$$\implies \alpha_i = \frac{Q_i - Q_m}{(R_m - Q_m) + Q_i - R_i}$$

If $\alpha_m \leq \alpha_i < 1$, then component $i$ becomes larger than component $m$ at $\overline{QR}(\alpha_i)$. We can compute this for all other indices (producing set $A$ in the algorithm) then find the first index that becomes larger than $m$. We assign this index to $m$ and its ratio to $\alpha$. If no such index exists, we can conclude that $m$ remains the maximum until $R$, thus additional endpoints are not needed.

Thus, within any two points in $P$ the maximum component stays the same, so the MaxPool can be exactly replaced with a linear map returning only that maximum element.    $\square$

At worst, then, for each of the $w$ windows each of size $s$, we may add $O(s)$ new endpoints ($\overline{QR}(\alpha)$ is monotonic in each component so the maximum index can only change $s$ times), and for each of those $O(s)$ new endpoints we must re-compute $D$, which requires $\text{O}(s)$ operations. Thus, the time complexity for each window is $\text{O}(s^2)$ and for the entire MaxPool computation is $\text{O}(ws^2)$.

Although most practical applications (especially on medium-sized networks) do not reach that worst-case bound, on extremely large (ImageNet-sized) networks we have found that such MaxPool computations end up taking the majority of the computation time. We believe this is an area for future work, perhaps using GPUs or other deep-learning hardware to perform the analysis.

## G   EXACTLINE for MaxPool + ReLU Layers

When a MaxPool layer is followed by a ReLU layer (or vice-versa), the preceding algorithm may include a large number of unnecessary points (for example, if the maximum index changes but

the actual value remains less than $0$, the ReLU layer will ensure that both pieces of the MaxPool output are mapped to the same constant value $0$). To avoid this, the MaxPool algorithm above can be modified to check before adding each $P_i$ whether the value at the maximum index is below $0$ and thus avoid adding unnecessary points. This can be made slightly more efficient by "skipping" straight to the first index where the value becomes positive, but overall the worst-case time complexity remains $O(ws^2)$.

## H    EXACTLINE for General Piecewise-Linear Layers

A more general algorithm can be devised for any piecewise-linear layer, as long as the input space can be partitioned into finitely-many (possibly unbounded) convex polytopes, where the function is affine within each one. For example, RELU fits this definition where the convex polytopes are the orthants. Once this has been established, then, we take *the union of the hyperplanes defining the faces of each convex polytope*. In the RELU example, each convex polytope defining the linear regions corresponds to a single orthant. Each orthant has an "H-representation" in the form $\{x \mid x_1 \leq 0 \wedge x_2 > 0 \wedge \ldots \wedge x_n \leq 0\}$, where we say the corresponding "hyperplanes defining the faces" of this polytope are $\{\{x \mid x_1 = 0\}, \ldots, \{x \mid x_n = 0\}\}$ (i.e., replacing the inequalities in the conjunction with equalities). Finally, given line segment $\overline{QR}$, we compare $Q$ and $R$ to each hyperplane individually; wherever $Q$ and $R$ lie on opposite sides of the hyperplane, we add the intersection point of the hyperplane with $\overline{QR}$. Sorting the resulting points gives us a valid $\mathcal{P}\big(f_{\restriction \overline{QR}}\big)$ tuple. If desired, the minimization described in Section 2 can be applied to recover the unique smallest $\mathcal{P}\big(f_{\restriction \overline{QR}}\big)$ tuple.

The intuition behind this algorithm is exactly the same as that behind the RELU algorithm; partition the line such that each resulting segment lies entirely within a single one of the polytopes. The further intuition here is that, if a point lies on a particular side of *all of the faces* defining the set of polytopes, then it must lie entirely within *a single one* of those polytopes (assuming the polytopes partition the input space).

Note that this algorithm can also be used to compute EXACTLINE for MAXPOOL layers, however, in comparison, the algorithm in Appendix F effectively adds two optimizations. First, the "search space" of possibly-intersected faces at any point is restricted to only the faces of the polytope that the last-added point resides in (minimizing redundancy and computation needed). Second, we always add the first (i.e., closest to $Q$) intersection found, so we do not have to sort the points at the end (we literally "follow the line"). Such function-specific optimizations tend to be beneficial when the partitioning of the input space is more complex (eg. MAXPOOL); for component-wise functions like RELU, the general algorithm presented above is extremely efficient.

## I    EXACTLINE Over Multiple Layers

Here we prove Theorem 3, which formalizes the intuition that we can solve EXACTLINE for an entire piecewise-linear network by solving it for all of the intermediate layers individually. Then, we use EXACTLINE on each layer in sequence, with each layer partitioning the input line segment further so that EXACTLINE on each latter layer can be computed on each of those partitions.

**Theorem 3.** *Given any piecewise-linear functions $f, g, h$ such that $f = h \circ g$ along with a line segment $\overline{QR}$ where $g(R) \neq g(Q)$ and $\mathcal{P}\big(g_{\restriction \overline{QR}}\big) = (P_1, P_2, \ldots, P_n)$ is* EXACTLINE *applied to $g$ over $\overline{QR}$, the following holds:*

$$\mathcal{P}\big(f_{\restriction \overline{QR}}\big) = \text{sorted} \left( \bigcup_{i=1}^{n-1} \left\{ P_i + \frac{y - g(P_i)}{g(P_{i+1}) - g(P_i)} \times (P_{i+1} - P_i) \mid y \in \mathcal{P}\big(h_{\restriction \overline{g(P_i)g(P_{i+1})}}\big) \right\} \right)$$

*where* sorted *returns a tuple of the points sorted by distance from $Q$.*

*Proof.* Consider any linear partition of $g$ defined by endpoints $(P_i, P_{i+1})$ of $\overline{QR}$. By the definition of $\mathcal{P}\big(g_{\restriction \overline{QR}}\big)$, there exists some affine map $A_i$ such that $g(x) = A_i(x)$ for any $x \in \overline{P_i P_{i+1}}$.

Now, consider $\mathcal{P}\big(h_{\restriction \overline{g(P_i)g(P_{i+1})}}\big) = (O_1^i = P_i, O_2^i, \ldots, O_m^i = P_{i+1})$. By the definition of $\mathcal{P}\big(h_{\restriction \overline{g(P_i)g(P_{i+1})}}\big)$, then, for any partition $\overline{O_j^i O_{j+1}^i}$, there exists some affine map $B_j^i$ such that $h(x) = B_j^i(x)$ for all $x \in \overline{O_j^i O_{j+1}^i}$.

Realizing that $O_j^i, O_{j+1}^i \in \overline{g(P_i)g(P_{i+1})}$ and that $\overline{P_i P_{i+1}}$ maps to $\overline{g(P_i)g(P_{i+1})}$ under $g$ (affineness of $g$ over $\overline{P_i P_{i+1}}$), and assuming $g(P_i) \neq g(P_{i+1})$ (i.e., $A_i$ is non-degenerate), there exist unique $I_j^i, I_{j+1}^i \in \overline{P_i P_{i+1}}$ such that $g(I_j^i) = O_j^i$ and $g(I_{j+1}^i) = O_{j+1}^i$. In particular, as affine maps retain ratios along lines, we have that:

$$I_j^i = P_i + \frac{O_j^i - g(P_i)}{g(P_{i+1}) - g(P_i)} \times (P_{i+1} - P_i)$$

And similar for $I_{j+1}^i$. (In the degenerate case, we can take $I_j^i = P_i, I_{j+1}^i = P_{i+1}$ to maintain the partitioning).

Now, we consider the line segment $\overline{I_j^i I_{j+1}^i} \subseteq \overline{P_i P_{i+1}}$. As it is a subset of $\overline{P_i P_{i+1}}$, all points $x \in \overline{I_j^i I_{j+1}^i}$ are transformed to $A_i(x) \in \overline{g(O_j^i)g(O_{j+1}^i)}$ by $g$. Thus, the application of $h$ to any such point $y = A_i(x) \in \overline{O_j^i O_{j+1}^i}$ is $B_j^i(y)$, and the composition $(B_j^i \circ A_i)(x)$ is an affine map taking points $x \in \overline{I_j^i I_{j+1}^i}$ to $f(x)$.

Finally, as the $O_j$s partition each $\overline{g(P_i)g(P_{i+1})}$ and each $\overline{P_i P_{i+1}}$ partitions $\overline{QR}$, and we picked $I_j^i$s to partition each $\overline{P_i P_{i+1}}$, the set of $I_j^i$s partitions $\overline{QR}$. $\qquad\square$

This theorem can be applied for each layer in a network, allowing us to identify a linear partitioning for the entire network with only linear partitioning algorithms for each individual layer.

## J   Constant Gradients with EXACTLINE

In Section 4, we relied on the fact that the gradients are constant within any linear partition given by $\mathcal{P}\big(\text{RELU}_{\restriction \overline{QR}}\big)$ computed with Equation 3. This fact was formalized by Theorem 4, which we prove below:

**Theorem 4.** *For any network $f$ with nonlinearities introduced only by ReLU functions and $\mathcal{P}\big(f_{\restriction \overline{QR}}\big) = (P_1, P_2, \ldots, P_n)$ computed according to Equation 3, the gradient of $f$ with respect to its input vector $x$, i.e. $\nabla f(x)$, is constant within each linear partition $\overline{P_i P_{i+1}}$.*

*Proof.* We first notice that the gradient of the entire network, when it is well-defined, is determined completely by the signs of the internal activations (as they control the action of the ReLU function).

Thus, as long as the signs of the internal activations are constant, the gradient will be constant as well.

Equation 4 in our paper identifies partitions where the signs of the internal activations are constant. Therefore, the gradient in each of those regions $\overline{P_i P_{i+1}}$ is also constant. $\qquad\square$

However, *in general*, for arbitrary $f$, it is possible that the action of $f$ may be affine over the line segment $\overline{P_i P_{i+1}}$ but not affine (or not describable by a single $A_i$) when considering points arbitrarily close to (but not lying on) $\overline{P_i P_{i+1}}$. In other words, the definition of $\mathcal{P}\big(f_{\restriction \overline{QR}}\big)$ as presented in our paper only requires that the *directional derivative* in the direction of $\overline{QR}$ is constant within each linear partition $\overline{P_i P_{i+1}}$, not the gradient more generally. A stronger definition of EXACTLINE could integrate such a requirement, but we present the weaker, more-general definition in the text for clarity of exposition.

However, as demonstrated in the above theorem, this stronger requirement *is* met by Equation 3, thus our exact computation of Integrated Gradients is correct.

# K Further EXACTLINE Implementation Details

We implemented our algorithms for computing $\mathcal{P}\big(f_{\restriction \overline{QR}}\big)$ in C++ using a gRPC server with Protobuf interface. This server can be called by a fully-fledged Python interface which allows one to define or import networks from ONNX [45] or ERAN [41] formats and compute $\mathcal{P}\big(f_{\restriction \overline{QR}}\big)$ for them. For the ACAS Xu experiments, we converted the ACAS Xu models provided in the ReluPlex [12] repository to ERAN format for analysis by our tool.

Internally, we represent $\mathcal{P}\big(f_{\restriction \overline{QR}}\big)$ by a vector of endpoints, each with a *ratio* along $\overline{QR}$ (i.e., $\alpha$ for the parameterization $\overline{QR}(\alpha) = Q + \alpha \times (R - Q)$), the layer at which the endpoint was introduced, and the corresponding post-image after applying $f$ (or however many layers have been applied so far).

On extremely large (ImageNet-sized) networks, storing the internal network activations corresponding to each of the thousands of endpoints requires significant memory usage (often hundreds of gigabytes), so our implementation sub-divides $\overline{QR}$ when necessary to control memory usage. However, for all tested networks, our implementation was extremely fast and could compute $\mathcal{P}\big(f_{\restriction \overline{QR}}\big)$ for all experiments in seconds.

# L Floating Point Computations

As with ReluPlex [12], we make use of floating-point computations in all of our implementations, meaning there may be some slight inaccuracies in our computations of each $P_i$. However, where float inaccuracies in ReluPlex correspond to hard-to-interpret errors relating to pivots of simplex tableaus, floating point errors in our algorithms are easier to interpret, corresponding to slight miscomputation in the exact position of each linear partition endpoint $P_i$. In practice, we have found that these errors are small and unlikely to cause meaningful issues with the use of EXACTLINE. With that said, improving accuracy of results while retaining performance is a major area of future work for both ReluPlex and EXACTLINE.

# M Future Work

Apart from the future work described previously, EXACTLINE itself can be further generalized. For example, while our algorithm is extremely fast (a number of seconds) on medium- and large-sized convolutional and feed-forward networks using ReLU non-linearities, it currently takes over an hour to execute on large ImageNet networks due to the presence of extremely large MaxPool layers. Scaling the algorithm and implementation (perhaps by using GPUs for verification, modifying the architecture of the network itself, or involving safe over-approximations) is an exciting focus for future work. Furthermore, we plan to investigate the use of safe linear over-approximations to commonly used non-piecewise-linear activation functions (such as $\tanh$) to analyze a wider variety of networks. Finally, we can generalize EXACTLINE to compute restrictions of networks to higher-dimensional input regions, which may allow the investigation of even more novel questions.