[Reviews · NeurIPS 2019]

Reviewer 1



Overall, the paper is well-written and presents an important novel contribution to the field in the ExactLine algorithm, although the description of the algorithm is a little unclear. The ACAS Xu example showcases the usefulness of such an algorithm in investigating and diagnosing problems with the decision boundary of a NN (e.g. the non-symmetric portions of the ACAS Xu boundary). The use of ExactLine to investigate inaccuracies in Integrated Gradient approximations is also an important contribution. Finally, the investigation of adversarial examples is also very interesting; further similar experiments using ExactLine could be a key component of developing a solid understanding of adversarial examples in the future.

Reviewer 2



Detailed comments: I have several comments to this work: 1. In general, the idea of this paper is novel and interesting, which provides a new way for human to understand the neural networks. Three applications for ExactLine are introduced and some interesting results are shown to investigate questions about the neural networks. 2. For the Integrated Gradients, experimental results are expected to prove whether the exact IG computed by equation (8) can generated better saliency maps than the approximate IG computed by equation (6). 3. Although the proposed method provided some new insights about the behavior of the neural networks, the application of the proposed method seems limited. The authors also admit that the results for the adversarial examples do not show whether reduced density of linear partitions contributes to robustness. Is there a way to utilize the proposed method to better distinguish the adversarial examples or better design more robust neural networks? More thinking about how to utilize the proposed method are expected.

Reviewer 3



The paper proposes an algorithm for computing the partition of a line into segments corresponding to different regions in a ReLU network. After explaining the algorithm, the authors present an application by studying the decision boundary of a network that determines what action an aircraft should take in order to avoid a collision with an intruder. In particular, they apply their algorithm to compute the output of the network on all points on a grid of lines (in polar coordinates) and show benefits over sampling the output at a discrete set of points. The authors then discuss another possible application of the algorithm, namely the exact computation of integrated gradients (IG). They analyze the efficacy of approximations of IG and discuss possible improvements to commonly used heuristics. Finally, the algorithm is used to analyze adversarial examples in neural networks. The authors observe that the original image and the perturbed adversarial image are often separated by many regions, a fact that they take as evidence against the "linear explanation" of adversarial examples. They also note that networks trained to be robust to adversarial perturbations seem to have significantly fewer linear partitions. Overall, the paper is well written (even though mathematical notation is sometimes confusing, see below). My main concern is that I find that the algorithm for ExactLine is simply an iterated application of the most straightforward way of computing segments along a line induced by ReLU activations. For this reason, it's hard to believe that no one has used a similar strategy before, and the algorithm does not seem like a substantial theoretical contribution. On the other hand, the different applications and empirical observations that the authors present may be useful for practitioners (I'm not sure). More comments: - The notation used in Theorem 1 should be explained: Q_i and R_i are i-th coordinates of Q and R, but this is not stated (and is also confusing because elsewhere P_i is the i-th point of the partition). - What is QR with \leftrightarrow in Theorem 1? - A better explanation of the ACAS Xu example could be helpful. Why does "one usually wishes to probe and visualize the recommendations of the network" (l.120)? - Integrated gradients: it seems surprising that "number of samples needed is relatively consistent within each network" (l.194). How were the averages in Table 2 computed? What was the variance? - Isn't the trapezoidal approximation of an integral always better than left and right sums? If so, ExactLine did not "provide novel insights into the behavior of neural networks" (l.209). - Some of the figures and tables are not very clear. For example, the different lines in Figure 4 should be explained.

[Author Response · NeurIPS 2019]

We thank the reviewers for their feedback, which will help improve writing clarity and notation consistency.

*[Reviewer #1]* **Algorithm Descriptions:** As suggested, we will replace the numerical example with an extended
discussion of recursive EXACTLINE application that improves upon Supplemental Section 3.
**Extending lines beyond the adversarial example in Figure 4:** We have found some images where the classification
changes multiple times close to the adversarial example indicating that the adversarial example lies in a "peninsula".
We can quantify the size of such peninsulas using EXACTLINE, which we can include in the paper.

*[Reviewer #2]* **Impact of exact IG on saliency maps:** Saliency maps produced using exact IG are noticeably more
accurate compared to those generated from approximate IG. We will include these in the paper and code release.
**Regarding limited applications of proposed method:** The paper already demonstrates the varied types of applications
for EXACTLINE, ranging from the first method for exact computation of IG to an empirical falsification of the Linear
Explanation for Adversarial Examples. Furthermore, using exact IG computed using EXACTLINE as the baseline,
better *non-uniform* sampling methods for approximate IG could be devised.
**Using EXACTLINE to detect adversarial images:** Initial experiments show that densities (as defined on line 232)
computed using random EXACTLINE "probes" around natural images differ significantly from those around adversarial
images. Thus, EXACTLINE could use such densities to distinguish between natural and adversarial images.

*[Reviewer #4]* **Theoretical significance of EXACTLINE algorithms:** The "most straightforward way" to approach
the algorithm would be that of [34], which requires exponential time to enumerate all possible orthants. A key
observation underlying our work is that we get a significantly faster algorithm (worst-case polynomial time for a
fixed number of layers) by restricting the input to be one dimensional (a line). This insight opens a new direction of
research for network-analysis tools: approaches in [34,12] are precise but exponential time, approaches such as [39] are
overapproximations but polynomial time; in contrast, we show that restricting the input dimensionality allows precise
*and* efficient algorithms. Furthermore, we believe that the efficient handling of MaxPool and MaxPool+ReLU is a
substantial theoretical contribution, which we will include in the main paper in lieu of the example in Section 2.
**Usefulness to practitioners:** After presenting our quantitative results to the authors of the IG paper, they have agreed to
switch to trapezoidal sampling in their implementation. Our falsification of the linearity hypothesis directly contributes
to the community's knowledge, and our initial observation about the relative-linearity of robust networks opens up
future work to developing a better understanding of adversarial examples. Furthermore, to enable practitioners to use
our EXACTLINE primitive, we provide a well-documented and tested, open-source library for computing EXACTLINE
in the form of a gRPC C++ server parallelized with the Intel TBB library with Python frontend. Our optimized
implementation of EXACTLINE can handle large ImageNet-scale networks without running out of memory.
**What is $\overleftrightarrow{QR}$ in Theorem 1?** The *unbounded* line incident to $Q$ and $R$ (which line *segment* $\overline{QR}$ lies in).
**Usefulness of visualizing ACAS Xu:** Visualizing their decision boundaries is a common way to understand networks
in practice. Prior approaches would use sampling when visualizing; e.g., Figure 7 in [12], and Figures 2, 8, 10, 12–15
in [5]. Use of EXACTLINE avoids the use of sampling along one dimension. Probing the behavior of networks could be
used to synthesize candidate whole-network specifications, which can then be verified by tools like [12,39].
**Regarding consistency of number of IG samples within a network:** Line 194 should actually read "the *density* of
samples needed is relatively consistent within each network"; i.e., number of samples divided by distance between
image and baseline. We will correct this sentence along with Table 2 in the final paper; we thank the reviewer for
bringing this to our attention. In fact, the variance of the number of samples is high, while the variance for density is low.
Standard deviation of the number of samples, eg., convsmall-left in Table 2 was approximately 77.6 vs. convsmall-trap
having 46.97. This high variance is primarily due to some images being further from the baseline than others and some
skew in the distribution of the densities. However, the quartiles for *density* for convsmall-left are 3.23/4.36/5.56 while
the corresponding values for convsmall-trap are 2.26/2.82/3.94. We will update Table 2 with such statistics. Note that
the suggestion on lines 195–198 can likewise be updated; viz., we can compute high-percentile density using a set of
training inputs, and use this density to compute the number of samples required for a given test image.
**Aren't trapezoidal approximations always better?** This is not true in general, and surprisingly not true in practice
either. Consider $\int_0^1 f(x)dx$ for $f(x) = 0$ when $x \in [0, 0.99]$ and $f(x) = 1$ otherwise. With two samples, the
left-approximation is 0, while the trapezoidal-approximation is 0.5. The true integral (0.01) is closer to the left-
approximation than the trapezoidal. In fact, for all models reported in Table 2, there were images for which trapezoidal
sampling needed *more* samples than left sampling to get below 5% error, indicating that left-approximation was
sometimes better than trapezoidal.
**Explanation of Figure 4:** Counting the number of vertical lines in the top and second-to-bottom lines of Figure 4 shows
how the density of the FGSM is higher (more non-linear) than that of the random direction for the same network (normal
training), falsifying the fundamental assumption behind the linear explanation of adversarial examples. Comparing the
number of vertical lines on all of the black (normally-trained) lines with that of their green (DiffAI-trained) counterparts,
we can see that the DiffAI model is significantly less-dense than the normal one. We will separate Figure 4 into two
figures and explain each individually.

[Meta-Review · NeurIPS 2019]

This paper proposes an efficient algorithm to compute a one-dimensional restriction of a deep ReLU network (which is a piece-wise affine function). The authors leverage this algorithm to study adversarial examples and the "integrated gradients" method. Reviewers found this work clearly written and easy to follow. Despite some concerns about the significance of the approach, the reviewer discussion and author rebuttal revealed a clear potential for future use of this technique, which will be useful to improve our understanding of large deep neural networks. The AC thus recommends acceptance of this work.